# "Falling between the cracks": Investigating the competing challenges experienced by professionals working with people who hoard

Emma Kaminskiy[1]*, Chase Staras[1¤], Stuart Brown[2], Sharon Morein-Zamir[1]

1 School of Psychology and Sport and Sensory Sciences and the Possessions and Hoarding Collective, Anglia Ruskin University, Cambridge, Cambridgeshire, United Kingdom, 2 Adult Early Help Team, Cambridgeshire County Council, St Ives, Cambridgeshire, United Kingdom

¤ Current address: School of Social Sciences, Nottingham Trent University, Nottingham, United Kingdom
* emma.kaminskiy@aru.ac.uk

## Abstract

Hoarding disorder is characterised by both the distress associated with discarding and the resulting accumulation of possessions that clutter the home environment, and represents a substantial public health and social problem, requiring an effective multi-agency response. Although a recognised psychiatric condition since 2013, hoarding disorder is under-recognised within mental health treatment provision, and evidence-based treatment pathways are lacking. This study aimed to explore multi-agency working in practice and similarities and differences in how hoarding support is perceived across a broad range of front-line professionals. 35 semi-structured inter-views representing a wide range of services from health, social care, housing, and the voluntary sector were conducted and analysed thematically. Three overarching themes are reported: Unique challenges of supporting people who hoard, Conflicting needs of client vs. needs of the organisation, Managing role boundaries and psychological tensions. Findings consistently highlight the challenges specific to these cases. However, they also point to a lack of consensus between professional groups in terms of perceived problems and risks to be addressed. Collaborating effectively with others to meet the psychological needs of clients while ensuring risk mitigation and addressing broader organisational, environmental, and community concerns was found to be a key challenge and associated with often going beyond designated role boundaries to support the person. Our findings point to a need for greater support and training of a diverse set of professionals in psychological models of people who hoard to enhance knowledge, awareness, and confidence of the psychological dimensions of hoarding disorder, and to support them in feeling more emotionally and practically prepared. The results indicate a need for greater integration of mental health measures in the assessment of hoarding cases, and to ensure adequate care pathways with dedicated psychological support along with assignment of case workers/coordination.

**Data availability statement:** Selected anonymised data is available via figshare at the following url https://doi.org/10.25411/aru.27201642.v2. Additional data requests may be sent to the Anglia Ruskin University Ethics committee via email (se-research@aru.ac.uk), or contact made to the corresponding author.

**Funding:** The author(s) received no specific funding for this work.

**Competing interests:** The authors have declared that no competing interests exist.

## Introduction

Hoarding of possessions occurs to varying degrees across the general population, with many individuals feeling at times that they have accumulated too many possessions and have difficulty discarding them, resulting in cluttered home environments [1]. When this becomes extreme, inducing chronic and considerably impaired functioning and quality of life, individuals can be diagnosed with Hoarding Disorder (HD), which affects 2.5% of the population [2]. Defined in the *Diagnostic and Statistical Manual of Mental Disorders* [3] as persistent difficulties discarding possessions regardless of actual value and due to a perceived need to save the items and intense emotional attachment to them, hoarding is characterised by both the distress associated with discarding and the resulting accumulation of possessions that clutter the home environment [4,5].

Hoarding disorder is associated with other psychiatric comorbidities and with early adverse childhood experiences and traumatic events in adulthood (including bereavement) [6,7]. Unlike other mental health conditions, access to early intervention is less likely for hoarding related issues. Clutter that takes over active living spaces may require decades of accumulation. Instead, a person's first interaction with services may be when compulsory enforcement action that relates to risk and safety of the home and wider community is necessary, or as part of accidental discovery by health professionals after e.g. a health crisis leading to emergency responses in the home. This can be related to reduced insight and awareness of the severity of their hoarding and implications on their surroundings, coupled with the highly stigmatising nature of the condition [8]. As such, first contact with services, where enforced clearance type activities may result, could in fact be the cause of more distress and are re-traumatising, resulting in increased distress and a worsening of the overall hoarding behaviours in the long term [9–12]. Related to this, people who hoard can have increased reticence and are mistrusting towards services, can be vulnerable, and isolated, and as such are often hard to reach and under studied [13,14]. Consequently, although an established psychiatric condition since 2013, HD is under-recognised with mental health treatment provision often lacking in practice.

Within the UK National Healthcare System (NHS) there are virtually no widely accepted evidence-based treatment pathways. The National Institute for Health and Care Excellent (NICE) guidelines for hoarding are presently covered under those for Obsessive Compulsive Disorder (OCD) and Body Dysmorphic Disorder [15]. In practice, support and service provision fall, for the most part, to local authorities (LAs) mandated under the Care Act [16] (see also [17]). LAs are expected to put forth multi-agency protocols for service provision including but not limited to social care, fire services, environmental health, first responders and housing. However, services and protocols across the UK appear inconsistent [12,18,19]. Moreover, a cursory inspection of publicly available LA protocols indicates they have similar yet distinct procedures, actors, and key partners. In North America and elsewhere task forces have been piloted to support multi-disciplinary and time sensitive coordinated care and have shown some promise [1,20–22].

Partnerships between agencies that allow for different priorities and perspectives towards hoarding to be shared is essential given the involvement of numerous services, and the need to mitigate wider risk to the individual and others, alongside a need for psychological treatment. Some multi-agency protocols have shown some success through increased coordination between diverse disciplines involved, from regulatory and enforcement agencies, housing, social care, and health [23]. Nonetheless, challenges of leadership while balancing the role with other primary tasks for coordinators have been reported [21].

There is limited research that has explored experiences of multi-disciplinary team (MDT) working for hoarding in the UK. Several studies have investigated distinct groups of service providers. For example, Porter and Hanson [24] explored the nature, circumstances and extent that hoarding presents for 11 housing officers working in on locality in England. This highlighted the need for more staff support, and an apparent gap in training support for front line professionals. Another study examined the understanding of self-neglect and hoarding in 44 adult safeguarding leads and managers [25]. This highlighted the finding that by the time safeguarding professionals are involved, a crisis response may be all that is offered, with the authors raising the key role of other agencies in providing greater early help. Another study interviewed 10 social workers who worked with individuals who hoard, reflecting on the at times contradictory challenges of having to navigate law and policy, organisational and multi-agency demands [26]. Haighton and colleagues [12] conducted 2 focus group interviews with professionals from housing, health and social care in the northeast of the UK, to explore possible collaborative team models and interventions. They found that there was a reliance on enforcement, and often a lack of shared understanding of how to support those needing help. Importantly, there was a consensus that multi-agency approaches are appropriate concurring with the broader literature [21] and the present policy framework in the UK.

Given the consensus on the need for multidisciplinary and multi-agency approaches to support individuals who hoard, this study aimed to explore MDT working in practice and the barriers and enablers to effective collaboration in supporting people who hoard. This study considers the experiences of a broad range of front-line professionals within a circumscribed region to allow for a better understanding of the experiences of different professional groups and the challenges they face. Further, this study explored similarities and differences in how hoarding support is perceived across front-line workers from diverse backgrounds.

## Materials and methods

Semi-structured interviews were conducted using a purposive sample of professionals who had attended a regular forum to discuss hoarding. This forum provides support and practical advice to professionals working with people who hoard across a large county in the East of England, serving both urban and rural communities.

### Ethics

This study received ethical approval from the School of Psychology and Sports Science at Anglia Ruskin University (ETH2223-3132). All participants provided informed written consent prior to participation.

### Sample and recruitment

The sample consisted of 35 professionals who work with people who hoard across a region in the East of England. Professionals from a range of backgrounds and sectors including, but not limited to: Disability Support; Adult Early Help; NHS Foundation Trusts; Housing Associations; Social Care; and Emergency Services. See Table 1 for a more detailed list of participant characteristics. The research was advertised to forum members in meetings and email communications. Participants who contacted the team received the information sheet and consent form. Those who provided informed written voluntary consent were invited to interview.

**Table 1. Participant characteristics.**

| Pseudonym | Gender | Job role | Description | Organisation/Sector |
|---|---|---|---|---|
| Sarah | Female | Community Outreach Officer | Works within an animal charity to support people with animal-related needs, e.g., neutering. | Voluntary Sector |
| Benjamin | Male | Regional Resident Support Manager | Helping people in crisis to access appropriate statutory services, including physical and mental health, addiction, benefits, domestic violence etc. | Housing Association |
| Chloe | Female | Housing Enforcement Team leader | Managing and supporting a team of people to visit homes and provide support to manage home conditions. | Council |
| Kelly | Female | Autism Support Practitioner | Supporting autistic people to manage day-to-day living. | Council |
| Evelyn | Female | Adult Support Coordinator | Managing adult social care referrals and arranging formal support. | Council |
| Jan | Female | Community Safety Officer | Home visits at the request of other professionals to give fire safety advice. | Fire Service |
| Lucy | Female | Named Nurse Safeguarding | Offering advice and guidance to GP practitioners. | NHS |
| Amanda | Female | Local Housing Manager | Helping people to safely manage their properties. | Housing Association |
| Madison | Female | Local Housing Manager | Looking after estates and tenants, including fire safety checks, antisocial behaviour, and state inspections. | Housing Association |
| Shirley | Female | Neighbourhood Response Officer | Ensuring condition of properties and helping vulnerable tenants maintain a habitable home condition. Preventing tenancy breaches. | Housing Association |
| Natasha | Female | Social Worker | Contracted social worker who works across multiple teams to support people in the community with a variety of complex needs. | Council |
| Kate | Female | Community Navigator Coordinator | Supporting people who can't manage at home through receiving referrals from family or statutory services and signposting them onto community resources and practical support. | Voluntary Sector |
| Karen | Female | Team Leader Community Wardens | Managing a team of wardens who help older people with practical tasks, e.g., collecting prescriptions. | Voluntary Sector |
| Verity | Female | Social Prescribing Link Worker | Having appointments with patients referred from GPs to establish their personal needs and connect them to appropriate support services. | NHS |
| Ron | Male | Deputy Manager Good Life Service | Helping people manage a range of mental health conditions. | Voluntary Sector |
| Paul | Male | Adult Support Coordinator | Supporting adults with learning disabilities through conducting care and support assessments and contacting appropriate services. | Council |
| Richard | Male | Senior Environmental Health Officer | Managing cleanliness and safety of properties in accordance with Environmental Health legislation. | Council |
| Yasmin | Female | Lead Reablement Worker | Assessing people's reablement needs and ensuring their access to appropriate support and services. | Council |
| Sadie | Female | Support Worker | Providing support for people who hoard and those with housing needs. | Council |
| Savannah | Female | Community Outreach Officer | Works within an animal charity to support people with animal-related needs, e.g., neutering. | Voluntary Sector |
| Emily | Female | Social Worker | Supporting adults to maintain their independence and manage day-to-day living. | Council |
| Maria | Female | Social Prescriber | Having appointments with patients referred from GPs to establish their personal needs and connect them to appropriate support services. | NHS |
| Blake | Male | Housing Enforcement Officer | Managing the safety or properties as well as the needs of tenants. | Council |
| Alice | Female | Social Prescribing Link Worker | Having appointments with patients referred from GPs to establish their personal needs and connect them to appropriate support services. | NHS |
| Stephen | Male | Environmental Health Protection Officer | Managing environmental nuisance complaints. | Council |

*(Continued)*

**Table 1.** (Continued)

| Pseudonym | Gender | Job role | Description | Organisation/Sector |
|---|---|---|---|---|
| Alesha | Female | Safeguarding Social Worker | Managing safe hospital discharges, checking property is safe to return to (I.e., navigating cleanliness, domestic abuse, self-neglect). | Council |
| Amelia | Female | Housing Support Worker | Supporting people in homeless shelters, i.e., shopping, money management, PIP applications etc. | Voluntary Sector |
| Dane | Male | Patient flow Coordinator | Support community team in managing vulnerable patients, e.g., elderly and house-bound, palliative care. | NHS |
| Patricia | Female | Help at Home Manager | Managing hospital discharge. Helping with practical elements where family and friends are not available. | Voluntary Sector |
| Saoirse | Female | Neighbourhood Response Officer | Ensuring condition of properties and helping vulnerable tenants maintain a habitable home condition. Preventing tenancy breaches. | Housing Association |
| Kira | Female | Reablement Lead | Assessing people's reablement needs and ensuring their access to appropriate support and services. | Council |
| Sasha | Female | Social Prescribing Link Worker | Having appointments with patients referred from GPs to establish their personal needs and connect them to appropriate support services. | NHS |
| Zara | Female | Adult Support Coordinator | Managing capacity assessments for adults with social needs and referring them to appropriate support. | Council |
| Grace | Female | Consultant Clinical Psychologist | Supports triaging and assessment for complex cases and offers trauma informed psychological support services | NHS |
| Lily | Female | CBT (high intensity) therapist | Offers CBT approaches as part of talking therapies solutions. | NHS |

## Data collection

Data was collected through semi-structured interviews, lasting 45 minutes on average (25–75 minutes). Data was collected between 20th February and 7th November 2023. All interviews were conducted by CS and EK who are experienced in qualitative research. Prior to commencement, participants were reminded of the research aims and their rights to withdraw.

Interviews were guided by an interview schedule developed based on previous literature and discussions with experts. The schedule covered background information on participant job roles, experiences of working with people who hoard and their role in the lives of people who hoard, how they navigate multi-disciplinary and multi-agency working, and what further support they would find beneficial. On completion of the interview, all participants were verbally debriefed and followed up by email, where a £10 Amazon voucher was provided. Interviews were conducted on Microsoft Teams, and both video and audio recorded, and transcribed verbatim. Transcripts were anonymised and pseudonyms were given.

## Data analysis

Transcripts were placed in an NVivo project file, and analysed using thematic analysis [27]. Transcripts were primarily coded by CS and checked by EK. During and following coding, the research team met repeatedly to discuss the findings and identify themes.

## Results

Data was obtained from 35 professionals working with people who hoard (see Table 1). Three themes were identified:

1. Unique challenges of supporting people who hoard.

2. Conflicting needs of client vs. needs of the organisation.

3. Managing role boundaries and psychological tensions.

## Theme 1 – Unique challenges of supporting people who hoard

Previous literature has illuminated the challenges encountered by professionals in both statutory services and the voluntary sector, such as staff overload and the management of performance targets [28,29]. These challenges are primarily rooted in funding and resource limitations. Practitioners in our sample further refer to specific challenges unique to professionals providing support for individuals, comprising sub-themes: a) Rising Demands and Complexity, and b) Differing Perceptions and Priorities.

**Rising demands and complexity.**  Practitioners note the increasing demands placed on professionals when supporting hoarding cases that transcend organization boundaries. Here the "complex social needs, mental health needs, and a history of trauma" (Sasha, Social Prescribing Link Worker, NHS) that often accompany hoarding clients and the specific emotional and psychological components that are difficult to train for:

*"They're complex problems as usually I would say in my very limited experience […] they're usually quite long term, perhaps psychological, emotional distress and disturbance over years. And that makes them quite complex. So it's not just a case of you know, tidying up some boxes and throwing out stuff. It's much more complex because it's to do with what all those things mean to somebody and so that's I think that's yeah, it's just complex really and emotional, psychological and you have to be very tactful and careful and have a lot of time. Time and complexity, I would say the two things". -* Verity, Social Prescribing Link Worker, NHS.

*"Every case is so unique, no one can teach you to deal with that"* - Chloe, Housing Enforcement Team Leader, Council.

Our present data also sheds light on the continuing intensification of hoarding cases from a professional's case-management perspective over time and more recent trends in increasing acuity. As Amanda, a Housing Association worker, puts it:

*"…the nature of the referrals and requests for help and support that come in are becoming more acute."* - Amanda, Care Housing Manager, Housing Association.

Participants emphasized that the current social and economic climate, including events like COVID-19 (e.g., using possessions to replace social interaction; *"It's just boxes and boxes and rooms full of Amazon boxes piled up"*; Jan, Community Safety Officer, Fire Service) and the Cost-of-Living Crisis (i.e., seeing all things as valuable), has further intensified the complexity of cases they handle as professionals. Participants from a variety of sectors reported on the challenges of helping people who hoard and increasing need:

*"I think the nature of the requests that are coming in is definitely more severe. Before, when I first started, it was people wanting signposting to practical support like gardening, housekeeping, shopping, and social activities. But certainly, since COVID […] they found that their support networks just disappeared overnight […] People are desperate for support."* - Kate, Community Navigator Coordinator, Voluntary Sector.

This increased need is being compounded by the perceived vulnerability of service users who hoard, where a reticence to seek help is seen as an additional burden to staff in terms of needing to be more proactive with communication and planning over a long period of time.

*"Hoarders, you don't hear from. We're the ones bugging them because you know they want to stay alone, they don't want to be bothered. They don't want the hoard cleared often. So actually, you know, with a hoarder case, you could quite happily sit there for six months and not do anything, and they wouldn't be on your back with their other work. You can't go six hours without being nagged, so it's very, very different. But that is why they're so vulnerable because*

*they're quiet. So actually, it makes them more vulnerable, because if we've got all these loud people shouting at us, they always will take priority and we have to remember you know the silent ones are just as risky and, you know, we need to focus on them as much, even though they don't want us there. So there's a big difference".* Chloe, Housing Enforcement Team Leader, Council.

**Differing perceptions and priorities.**  Nevertheless, while professionals share the overarching pressure, the manifestations of these challenges vary significantly in terms of perceived problems and risks to be addressed. For example, housing professionals are primarily concerned with property conditions, mental health practitioners focus on well-being, home helpers/nurses deal with tasks such as wound care and meal preparation, and environmental health experts and the fire service prioritize pest control and fire safety. Moreover, client capacity and cooperation are often complicated by a lack of trust, at times limited insight and the highly stigmatising nature of the condition leading to feelings of shame and reticence.

**Housing and fire:**

*"The kitchens are where most fires start. So that's a real focus for the fire service, the fire service focus on two things, means of escape and the condition of the kitchen and if we can get back in again, I think that's what I would measure success […] Getting a property cleared, evicting somebody. I don't see those as they're not long-term goals with serious hoarders"* - Madison, Local Housing Manager, Housing Association.

**Environmental health:**

*"We're committed to getting it to that point where we're satisfied. That doesn't mean necessarily a full house clearance […] if there ain't a public health risk, you know, then we would walk away"* - Richard, Senior Environmental Health Officer, Council.

**Home helpers/nurses:**

*"There is a big difference I think to social workers, you know, social care and perhaps health because health care … which I understand because you're going in, you're probably doing leg dressings, you want a sterile environment. And you're probably having to kneel down to do people's dressings on their limbs. So therefore, you don't want to kneel on a very squelchy carpet"* - Kate, Community Navigator Coordinator, Voluntary Sector.

**Psychology and mental health practitioners:**

*"So you might have a range of tools you use but they have to apply that to that area…with people who hoard… we can't visit people in their own homes, we can maybe get them to bring something in that they don't know what to do with, so you know, we can get them to do some practical work but we can't really ascertain the level of clutter in people's homes, and also the way people hoard is very very different"* – Lily, CBT therapist, NHS.

Often, any needs that are not met by these sectors are passed forward to social care teams; common given that cases are often passed between sectors and where responsibilities and priorities of each organisation are not distinct:

**Social care:**

*"And as I say, lack of services […] so other professionals might walk in and go "oh God. This is terrible. You really need social care support, social care, you need to do something about it", but going back to capacity, that person can choose how they live, we can't do anything about it, so I think it's that unrealistic expectation of what social care can do"* - Natasha, Social Worker, Council.

This diversity can complicate the establishment of priorities and protocols when working with individuals who hoard and the development of a joint approach, agreeing on the primary concern and main stakeholders to be involved. Thus, while the severity of hoarding cases has increased across all organizations, the way professionals experience these challenges, the perceived presenting problems and risks, varies. Complicating matters further is the uniqueness of each hoarding case (see above), making it essential to establish distinct priorities and action plans for every client.

### Theme 2 – Conflicting needs of clients vs needs of the organisation

Participants in this study reflected on the intricate task of managing the often-conflicting needs of clients while navigating their roles within the wider organizational, environmental, and community contexts. Professionals found themselves grappling with dilemmas regarding which issues to prioritize and how to collaborate effectively with others to meet the psychological needs of clients while ensuring risk mitigation and addressing broader concerns. Three sub-themes are presented: a) Client needs vs. wider community b) Client capacity and wants vs risk to professionals, and c) Client behaviours vs family members' needs.

**Client needs vs. wider community.** Professionals frequently find themselves at the crossroads of managing clients' issues (e.g., clutter and unsafe living conditions) while also addressing the clients' mental health needs and willingness to engage with service providers. Moreover, client-centric needs have to be delicately balanced with the responsibilities imposed by their respective organizations. For instance, Benjamin, a Regional Resident Support Manager within a housing association, sheds light on the complexities of this balancing act:

*"At the same time, we have a duty of care as a landlord to make sure we've got a property that's maintained and is in good condition. And if that then breaches those, then it's the tenancy sort of issue, then it becomes a problem. So, then there's barriers if you like in terms of what we can do. You know, I can't do the best work if they're not willing to engage, there's no support services that can help the individual, engage the individual. Then we go down the enforcement route and then you know, then we're left with a position of stalemate where you know, effectively, the resident could be forced out of their home, yeah. And what will they do? They will leave their property, abandon their property and then start all over again"* - Benjamin, Regional Resident Support Manager, Housing Association.

This balancing act is a vivid illustration of how professionals face competing responsibilities, not just to the hoarding individual (and the associated issues with navigating a lack of support to "help the individual engage"), but also to the specific organisation and wider community. Neglected property conditions can adversely affect neighbours, leading to concerns about vermin infestation and structural damage, which only exacerbates the ongoing challenges posed by hoarding behaviours. These wider environmental concerns distinguish hoarding clients from broader mental health cases where failure to adequately address the problem has considerably wider implications beyond the individual.

Conversely, mental health professionals in our sample, who are trained in psychological models and frameworks expressed the perceived gap in not having a direct understanding of the home environment and are not connected directly to other teams involved, and therefore feel unable to walk alongside to support people in the decluttering activities that might be needed. Here Grace discusses the need for more support and to be more adaptive to working with others on graded exposure, and de-cluttering of homes, activities which are often overwhelming for clients to embark upon:

*"It was just so overwhelming for her, the whole house was just full of stuff and just going through it and, so, our support workers do do some graded exposure with people who are anxious … because we don't expect those people to be able to just suddenly walk to the shop or get on the bus, we do that with them. But then, actually, who does that bit with the people who hoard? …I could do the work on really understanding what's happening and share a formulation with*

*someone and they could do a graded exposure plan but there often needs to be a bit longer and practise strategies you've learned and that's quite difficult because like, you know, someone telling you the exact diet or exercise plan you have. But then you also need a personal trainer you know, because it's so overwhelming a task"* – Grace, Consultant Clinical Psychologist, NHS.

**Client capacity and wants vs. risk to professionals.** In a similar vein, professionals are further tasked with navigating the needs of their clients while ensuring their own safety. Hoarded properties present hazards such as vermin infestation or structural integrity compromised by hoarding. As Karen, a Team Leader in the Voluntary Sector, pointed out:

*"Yes, so from a care perspective, when you're delivering hands-on care that's been provisioned by the social services, it's obviously it's incredibly challenging because […] people don't have a lot of insight into their disordered behaviour, if you like. It's people's homes so we've got to respect how people want to live. But at the same time, it's our workplace. So then you've got the sort of health and safety in the workplace issues like coming to play"* - Karen, Team Leader Community Wardens, Voluntary Sector.

It is important to note that whilst professionals must carefully navigate issues of capacity and consent when assisting individuals with hoarding tendencies and conducting property clearance, workplace health and safety standards impose different requirements that must also be met. Furthermore, several of these sectors do not have hoarding case management explicitly outlined in their roles or responsibilities, leaving them without specific guidance on how to address these challenges and therefore present additional psychological tension for front-line staff (see also theme 3).

**Client behaviours vs family members' needs.** While professionals navigate the home environments of clients, they may also encounter challenges arising from additional family members residing in the hoarded property. These family members may have their own health needs which are exacerbated by the hoarding conditions within the home. Staff may feel they are unable to adequately address issues without taking into account other family members in the home, which conversely they may feel is outside of their remit. Karen provided an example:

*"There was one case we went into where this lady, incredibly vulnerable lady. She had dementia, very poor mobility. Needed full care, you know. And she lived with her. Well, her daughter lived with her. So, it was the lady's house. And the daughter lived, and it, the hoard, was the daughters. She was a massive problem for her and to the point where you know, it was affecting this lady's house. She couldn't. She couldn't. You couldn't fit a bed in the room. So, the lady couldn't sleep in a bed. She had to sleep in a chair, and this was causing all sorts of problems. Health problems with her legs and leg ulcers and things like that. And in that circumstance and to be fair to the daughter she was aware of this, and she said mum needs her feet up. I can't, you know, but I need to get her recliner chair, but I can't"* Karen, Team Leader Community Wardens, Voluntary Sector.

Thus, there are specific challenges of working with people who hoard that are distinct from service sector limitations (i.e., scarce capacity and resources) centred around establishing key priorities for working with people who hoard. Alongside this, professionals need to balance client needs and capacity with wider responsibilities and concerns and therefore all decisions are a compromise where the ideal course of action can be *"like shifting tectonic plates"* (Alice, Social Prescribing Link Worker, NHS). This means there are multiple things at any one time that need to be considered by a professional working with people who hoard. This multifaceted challenge of managing hoarding varies across professional groups and underscores the complexity of establishing clear priorities, and the necessity of multi-agency collaboration and external support; the challenges and consequences of which are discussed in the next theme.

### Theme 3 – Managing role boundaries and psychological consequences

The multifaceted nature of hoarding cases, coupled with the varying challenges faced by different sectors, makes it difficult to establish clear priorities. This lack of cohesion often results in professionals taking on responsibilities that go beyond their designated role boundaries, leading to tensions and psychological distress when trying to support clients whilst attempting collaboration with other services. The theme comprises a) A culture of referrals and the consequences for the voluntary sector, b) The need for psychological understandings and support, and c) Consequences for professionals: Going above and beyond and professional blagging.

**A culture of referrals and the consequences for the voluntary sector.** Professionals in our study report tensions arising from the complex interplay of responsibilities among different sectors, which could lead to frustration, a sense of helplessness, and poor MDT working. These tensions contribute to the protracted nature of hoarding cases and a lack of ownership and responsibility from any single service. Dane, a Patient Flow Coordinator, illustrates the complexity of navigating across different services:

*"I said it sort of falls between this crack between three different services. You know, social care, housing and health. And it just seems to fall. It doesn't fall between because of the nature of the it, it falls between all three of us and so occasionally you can feel like it's being bounced around and not getting done because nobody wants to take it on because it's particularly complex and laborious or they have an objection because in their professional opinion, it's not their responsibility, it's yours. And in our professional opinion, it's not our responsibility. It's housing. And so, it can feel like it just goes round and round a bit until like because somebody please take responsibility for this patient, you know it's not getting any better sooner we get something in there".* - Dane, Patient Flow Coordinator, NHS.

Alice, extends these understandings:

*"And we understand about capacity in everyone's capacity is quite stretched, but it's like sometimes you just want to know where can you go with this because you're concerned about this individual human being. But sometimes when they're not known to someone. You know, so you get to know this person. There's this human in front of you you're worried about. But when it's just a name, and do they tick these boxes? It's it's easier. For all we know, it's not our roles our you know it's not. They don't meet these criteria. So therefore, according to the legislative framework, but actually no one's black and white [researcher: yeah]. Everybody's quite complex and you need to work a bit more creatively sometimes to be able to get them the help they need [researcher: Yeah]. So yeah, it can. It can. It can cause some difficulties in working with those agencies".* Alice, Social Prescribing Link Worker, NHS.

Consequently, the voluntary sector often bears the brunt of these multi-agency issues, further complicating the collaborative effort. Kate, a Community Navigator Coordinator from the Voluntary Sector, expressed her frustration with the prevailing culture of signposting and referrals:

*"I think you know there is a lot of pressure on professionals to signpost to something else [researcher: Yeah]. And often it is to the voluntary sector. And I find this quite frustrating at times that instead of, you know working in a multi-agency approach that it is very much like ohh it's not our responsibility."* - Kate, Community Navigator Coordinator, Voluntary Sector.

Kate, highlights not only the increased burden being placed on the voluntary sector because of increasing demand and lack of dedicated pathways for hoarding cases, but also the perceived marginalisation of their role within wider

collaborative team efforts, with statutory and health sectors dominating decision-making processes despite the extensive role those from the voluntary sector play in hoarding cases:

*"I find it quite difficult cause I do find representing the voluntary sector you don't have an equal participation in some of these groups. You know, it is very much taken over by statutory and health. And yeah, I do think sometimes our role is marginalised".* - Kate, Community Navigator, Voluntary Sector.

**The need for increased psychological understandings and support.** Most people interviewed highlighted the importance of providing mental health support, but some reported a sense that the necessary psychological support services are often lacking. Benjamin raised the issue of a noticeable gap in mental health provision:

*"Other support services are required. Whether that be as I say, mental health or whatever. You know, the reality is within the mental health team, there's not anyone. And there's not a team that that deals with hoarding, you know …. And again, that's nationwide, but I don't think that's, you know uncommon but obviously it is a disorder, and it is you know it does need help and assistance and there just seems to be a lack of funding available for this type of work that's required".* - Benjamin, Regional Resident Support Manager, Housing Association.

Participants in our study often reported that they did not feel equipped with the skills necessary to manage hoarding cases. As noted by Karen:

*"Unless you are properly a mental health professional, you can't […] you would need to do a psychology degree I think to properly understand it".* - Team Leader Community Wardens, Voluntary Sector.

For others the need for more training and support in specialist evidence based psychological therapies and strategies was paramount to improve the confidence and skills of practitioners when supporting people who hoard:

*"It's training needs, just knowing from experts what are the current psychological models that people draw on, different strategies that people use when they are going into people's homes and de cluttering… how you might adapt things"* – Grace, Consultant Clinical Psychologist, NHS.

*"It's like a lot of in our sort of area of work, any sort of support area or psychological therapies or any like that. There's quite a bit about understanding the condition or assessing the need [researcher: Mm-hmm]. But there's there's not so much on, so what can we do to help this person [researcher: yeah]. And I feel like with the hoarding, in the whole area of hoarding, that's a bit of a gap that would be good yeah. Not so much about what can we do practically, but what can we do for that person in other ways that would help them to manage themselves a bit better. So they are then in a better place to accept the help, the practical help that you can give. That's I think where there is a little bit of a gap maybe"* - Alice, Social prescriber GP surgery, NHS.

**Consequences for professionals: Going above and beyond and professional blagging.** The perceived dearth of practical support often compels professionals to operate beyond their designated roles, leading to what some participants referred to as "*professional blagging*" where professionals find themselves "*blagging each case and hoping for the best*" (Chloe, Housing Enforcement Team Leader, Council) as they take on extended responsibilities outside their remit. Sasha, a Social Prescribing Link Worker, confessed to going beyond her role boundaries due to resource limitations:

*"I get in trouble for doing stuff that I shouldn't be doing so. Yeah. I mean, really, my role should be to find the resources for the person and direct them to the resources and maybe get some funding and get some outsource for some help.*

*But because that help isn't there, I often end up going round and physically, practically helping myself and that's outside of my remit for my role"* - Sasha, Social Prescribing Link Worker, NHS.

Kate, from the voluntary sector, shared a similar experience:

*"Last year we couldn't get in. The ambulance crew couldn't get into somebody's house. So social prescriber and I went with bin bags and rubber gloves, and we managed to prise the door open and there was. Well, we filled 10 black bin liners with rotting food because Tesco's had just done deliveries and literally put the stuff in and everything was like yoghurts. It was milk that had just so yeah, so we do, we do roll our sleeves up. I don't, I think my manager would be absolutely horrified cause this is not really part of our job"* - Kate, Community Navigator Coordinator, Voluntary Sector.

The majority of participants in our study reflect on the emotional impact this has and the need for more psychological self-care support for staff:

*"It gets you. I do take stuff home with me, so it's. No, I don't always feel prepared for what I'm going to face. I can't just switch it off and go to sleep at nighttime. Stuff does bother me you know. I think about things when you're lying in bed at 3:00 O'clock in the morning, you're like because you can't settle, you know, and you're thinking of how somebody's living, how they've gotten like that. How can I fix it? What can I do? I understand it's a mental health condition going back to the hoarding part, the hoarders. But it also affects our mental health as well, you know, because you you're worrying, you know, you're stressed about it […] there's a sink or swim situation in most aspects of the job. You know, from hoarders to people threatening me, they're going to stab me. I've been trapped in kitchens"* - Saoirse, Neighbourhood Response Officer, Housing Association.

*"You know, cause sometimes I think, you know, the staff themselves need support. You know, this is very difficult. I mean when we have the MDT meetings, I think for any of us to walk alongside someone who is going through possibly having made the decision that they will try and change the way that they live is very time consuming. It's very demanding on them. It's quite demanding on the person who's supporting them. And often I think you know that the person who's trying to support them needs having that kind of debrief or extra support from someone else you know. It is very, very challenging and it's often very, very long term work"* - Kate, Community Navigator Coordinator, Voluntary Sector.

This emotional burden over often protracted periods of time with the additional strain of navigating unclear referral pathways highlights a need for more dedicated support and training, and for improved knowledge sharing and psychological self-care for staff.

## Discussion

The findings reveal a complex landscape of competing understandings and tensions surrounding hoarding. Our findings emphasize and support the multidimensional nature of hoarding: although hoarding is a recognised psychiatric condition, its impact goes far beyond the individual or a mental health focused framework. Professionals in our study represented a diverse set of agencies and even separate roles within the same agency. These frontline workers frequently found themselves at the crossroads of managing clients' social issues (e.g., clutter and unsafe living conditions) while also addressing the clients' mental health needs and willingness to engage with service providers.

Similar to previous findings, practitioners valued multi-agency collaborative case management approaches [12]. However, in practice, there was a lack of shared understanding about what to prioritise and how to navigate professional boundaries and work effectively with other agencies. This underscores the challenges that professionals encounter in providing holistic support and points to particular challenges of multiagency collaborative case management approaches

for hoarding disorder. Our study highlights the strain placed on MDT working [18], indicating a culture of referrals and signposting, constrained boundaries and overreliance on the voluntary sector. This impedes the facilitation of navigating appropriate support based on individual needs for people struggling with hoarding behaviours. MDT challenges and broader issues identified in the UK legislative context have been discussed elsewhere [30]. Safeguarding protocols and navigating The Mental Capacity Act [31] and the Care Act [16] have been criticised as being confusing for front-line workers, and, in practice, involving increased reliance on Social Care agencies for assessment processes. It has been proposed that the notion of capacity has been skewed in practice, often serving as a tool for agenda-led decision-making rather than ensuring comprehensive service provision for vulnerable groups [29,30]). In this way, hoarding is viewed as a lifestyle choice and problems of hoarding are framed within a social constructionist framework and do not in and of themselves require intervention [32]. This is at odds with biopsychosocial understandings which allow for inclusion of the individual's psychological and emotional components and an appreciation of hoarding as a maladaptive behaviour [33]. This may include distress associated with both the onset of the condition, as well as the impact living in cluttered environments may have on psychological and physical health. It may also include an appreciation that compulsively buying or avoiding discarding, are not necessarily by choice but as a way to overcome distress or as a coping mechanism for deeper psychological attachment issues and maladaptive social and interpersonal strategies [25,34].

While the individuality of each hoarding client has been identified in prior research (e.g., [12,24]), our participants emphasized that these challenges are not static and that the current social and economic climate, including events like COVID-19 and the Cost-of-Living Crisis (i.e., seeing all things as valuable), has further intensified the complexity of cases they handle as professionals. Professionals need to balance immediate risks, compliance and enforcement concerns on the one hand, with the client's mental health concerns and their engagement with services on the other. This, coupled with a perceived increased burden and complexity of cases for professionals presents unique challenges including tensions about perceptions and understandings of the problem and priorities to be addressed. For example, on the one hand, participants in our interviews recognised the psychological factors that contribute to hoarding, both in terms of understanding the underlying distress and psychological factors involved in onset and maintenance (e.g., trauma as its relationship with accumulation of possessions (e.g., [34]) and how cluttered and sometimes unsafe living environments impact on the persons functioning and psychological wellbeing [35,36]. On the other hand, problems were often formulated as relating to the person's environment or risk (both to the individual at work, the wider family, and the environment and clearing clutter). This could be attributed, at least in part, to the immediate required measurable outcomes sought (e.g., reducing a fire risk in a kitchen) compared to the long-term, intractable and difficult to assess aspects of mental ill-health and distress.

Professionals in our study highlighted the need for more awareness of and access to existing evidence-based non-specialist psychological support strategies for hoarding. Participants in interviews pointed to a need for more dedicated training and an improved shared understanding between teams and agencies about existing interventions and on the psychological factors involved, as well as protocols for where to find additional specialist psychological input to cases. An implication of our findings is that there is a need for additional awareness and training in social and psychological support for hoarding [36,37]. Participants in interviews often discussed using the Clutter Rating Scale [38], a widely known tool often featured in multi-agency guidelines. This is a particularly valuable validated and standardised assessment measure of hoarding severity which is easy to use and doesn't rely on expertise knowledge or interpretation of symptoms. However, this tool only considers the physical condition of the property and how much free space is available in a room, as opposed to a person's relationship with their possessions, levels of distress, or other factors associated with the collection of items (e.g., how ordered accumulated items are). Additional tools such as the hoarding rating scale [39] were not widely mentioned. Though requiring some training, the inclusion of such tools in the future may complement the use of the clutter image rating scale. Similarly, many professionals referred to not being aware of dedicated evidence-based therapies to support and help navigate the mental health aspects involved in hoarding. For example, Buried in Treasures [40] is a widely recognized, standardized intervention proven effective in various settings, including as a structured support group

format [41]. It serves as both a valuable resource for practitioners and a self-help tool and could be more fully leveraged. Instead, practitioners would often informally adapt existing non-hoarding protocols for supporting people or engage in professional blagging (see theme 3). In short, our findings suggest that existing protocols and practices being adopted within MDTs do not sufficiently address the complex mental health needs or offer a dedicated model of hoarding as a psychological issue [36,37].

Our findings point to the importance of providing clear or dedicated care pathways to support people who hoard, with evidence based psychological approaches embedded in best practice guidelines for MDT working. In light of these findings, there appears a need for greater integration of mental health measures in the assessment of hoarding cases, and to ensure adequate care pathways with dedicated psychological support along with the assignment of case workers/coordination. This would allow appropriate facilitation and identification of the complex needs and support required. This necessitates a shift towards a more holistic approach that acknowledges the psychological and emotional components underlying hoarding behaviours. Moreover, establishing clear priorities within each service sector and fostering coordinated care pathways are crucial steps toward enhancing multi-agency collaboration and reducing friction in the provision of support for hoarding cases (e.g., [42,43]). It may be that multiple concurrent interventions, in particular incorporating safety and harm reduction approaches are required in order to improve outcomes for people who hoard [44].

There are limitations to our study. The recruitment of practitioners relied on advertisement through a regional hoarding forum and may have inadvertently excluded perspectives from professionals who did not have access to or who chose not to participate and therefore some perspectives may have been missed. Recruitment focused on a wide region in the East of England, raising concerns that findings may not necessarily reflect or represent the wider UK context or those found elsewhere. However, from inspection of existent literature including grey literature and additional sources, it appears that in fact many of these issues do appear to generalise more broadly where multidisciplinary and multi-agency approaches have been adopted. A strength of this study is that the sample represents a broad range of professionals from different sectors. However, although diverse in its representation of professional groups and different sectors, the sample may not encapsulate all stakeholders involved in supporting people who hoard. Nevertheless, the present study encompassed the broadest range to date of professionals across services and, to an extent, within services as well.

## Recommendations and conclusion

Our findings support the wider call for the creation of clear and dedicated care pathways to support effective multi-agency collaboration. In addition, our findings suggest that there is a need to provide professionals with increased specialised mental health training to enhance knowledge, awareness and confidence of the psychological dimensions of hoarding disorder, and to support professionals to feel more emotionally and practically prepared in supporting people who hoard. Future research should embed psychological models to assess and support people who hoard and consider frameworks and protocols to support greater involvement from mental health services. Likewise, implementing complementary simple psychological assessments alongside the Clutter Rating Scale would provide a more comprehensive evaluation of hoarding cases, enabling a more person-centred understanding of the underlying psychological distress and its relationship to hoarding behaviours.

## Supporting information

**S1 File. COREQ checklist.**
(PDF)

## Acknowledgments

We would like to thank Faye Gentile, Student and Research Assistant, ARU, for her help in setting up the study and supporting the development of materials used in the interviews.

## Author contributions

**Conceptualization:** Emma Kaminskiy, Sharon Morein-Zamir.

**Data curation:** Emma Kaminskiy, Chase Staras, Sharon Morein-Zamir.

**Formal analysis:** Emma Kaminskiy, Chase Staras.

**Methodology:** Emma Kaminskiy.

**Project administration:** Chase Staras.

**Supervision:** Stuart Brown.

**Validation:** Stuart Brown.

**Writing – original draft:** Emma Kaminskiy, Chase Staras, Sharon Morein-Zamir.

**Writing – review & editing:** Emma Kaminskiy, Stuart Brown, Sharon Morein-Zamir.

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
