## [Decision Letter · Decision Letter 0]

6 Sep 2024

PONE-D-23-41611“Falling between the cracks”: Investigating the competing challenges experienced by professionals working with people who hoard.PLOS ONE

Dear Dr. Kaminskiy,

Thank you for submitting your manuscript to PLOS ONE. After careful consideration, we feel that it has merit but does not fully meet PLOS ONE’s publication criteria as it currently stands. Therefore, we invite you to submit a revised version of the manuscript that addresses the points raised during the review process.

We look forward to receiving your revised manuscript.

Kind regards,

Vanessa Carels

Staff Editor

PLOS ONE

Journal Requirements: 1. When submitting your revision, we need you to address these additional requirements. Please ensure that your manuscript meets PLOS ONE's style requirements, including those for file naming. The PLOS ONE style templates can be found at https://journals.plos.org/plosone/s/file?id=wjVg/PLOSOne_formatting_sample_main_body.pdf and https://journals.plos.org/plosone/s/file?id=ba62/PLOSOne_formatting_sample_title_authors_affiliations.pdf. 2. We note that you have indicated that there are restrictions to data sharing for this study. PLOS only allows data to be available upon request if there are legal or ethical restrictions on sharing data publicly. For more information on unacceptable data access restrictions, please see http://journals.plos.org/plosone/s/data-availability#loc-unacceptable-data-access-restrictions.  Before we proceed with your manuscript, please address the following prompts: a) If there are ethical or legal restrictions on sharing a de-identified data set, please explain them in detail (e.g., data contain potentially identifying or sensitive patient information, data are owned by a third-party organization, etc.) and who has imposed them (e.g., a Research Ethics Committee or Institutional Review Board, etc.). Please also provide contact information for a data access committee, ethics committee, or other institutional body to which data requests may be sent. b) If there are no restrictions, please upload the minimal anonymized data set necessary to replicate your study findings to a stable, public repository and provide us with the relevant URLs, DOIs, or accession numbers. For a list of recommended repositories, please see https://journals.plos.org/plosone/s/recommended-repositories. You also have the option of uploading the data as Supporting Information files, but we would recommend depositing data directly to a data repository if possible. We will update your Data Availability statement on your behalf to reflect the information you provide. 3. In the online submission form, you indicated that [Excerpts of the interview transcripts are included in the findings presented in the paper. However, there are restrictions on access to full interview transcripts, due to issues of confidentiality and anonymity for ethical conduct of qualitative research. The data contains potentially identifying and sensitive personal information and the authors do not have ethical permission to share full interview transcripts as standard. Data requests may be sent to the Anglia Ruskin University Ethics committee, or contact made to the corresponding author.]. All PLOS journals now require all data underlying the findings described in their manuscript to be freely available to other researchers, either 1. In a public repository, 2. Within the manuscript itself, or 3. Uploaded as supplementary information.This policy applies to all data except where public deposition would breach compliance with the protocol approved by your research ethics board. If your data cannot be made publicly available for ethical or legal reasons (e.g., public availability would compromise patient privacy), please explain your reasons on resubmission and your exemption request will be escalated for approval.  4. Please amend your authorship list in your manuscript file to include author Emma Kaminskiy, Chase Staras, Stuart Brown and Sharon Morein.  5. Please amend your list of authors on the manuscript to ensure that each author is linked to an affiliation. Authors’ affiliations should reflect the institution where the work was done (if authors moved subsequently, you can also list the new affiliation stating “current affiliation:….” as necessary).

Reviewers' comments:

Reviewer's Responses to Questions

**Comments to the Author**

1. Is the manuscript technically sound, and do the data support the conclusions?

Reviewer #1: Yes

Reviewer #2: Yes

Reviewer #3: Yes

2. Has the statistical analysis been performed appropriately and rigorously? 

Reviewer #1: N/A

Reviewer #2: N/A

Reviewer #3: I Don't Know

3. Have the authors made all data underlying the findings in their manuscript fully available?

Reviewer #1: No

Reviewer #2: Yes

Reviewer #3: Yes

4. Is the manuscript presented in an intelligible fashion and written in standard English?

Reviewer #1: Yes

Reviewer #2: Yes

Reviewer #3: Yes

5. Review Comments to the Author

Reviewer #1: This is a timely publication calling the attention of the medical and mainly the helper communities to the problem of hoarding disorder, as well as the unresolved isseue of its management. Albeit it is said to be prevalent (around 2% of the general population, more than epilepsy, schizophrenia, autism spectrum disorder and others), there is relatively few literature about it (Pubmed giving 2300 matches, in contrast to several ten-thousands of those above). The literature about its treatment is even more sparse; actually, hardly any (see for instance the excellent review:

Nakao, T. and Kanba, S. (2019), Pathophysiology and treatment of hoarding disorder. Psychiatry Clin. Neurosci., 73: 370-375. https://doi.org/10.1111/pcn.12853 ); showing our helplessness in managing this problem and treating those experiencing the disorder; the main difficulty being the lack of insight those affected.

There is a lack by those with hoarding disorder recognizing the pathologic nature of their "collecting passion" (reaching sometimes the degree of lack of capacity), versus a contradictory nature of ihelpers' interveining into private lives of people with dubious diagnoses, and having in some cases traumatic backgrounds; versus individual and public needs of safety; raising the need of consideration and possibly, legal approach together with harmonizing the multidisciplinary cooperation.

The authors face and show these problems categorizing them into thematic groups, and, in a big work, elaborating 35 unstructured interviews (lasting for an average of 45mins) with those helpers- volunteers or statutory employees. This great effort is impressing, and at the same time, giving a real-word approach of the difficulties tackling the disorder. The authors do not aim to deal with the controversial disorder itself, but through the interviews, one may have some insight into it as well, allowing to see the problem in its complexity. It is written in excellent English (based on my humbe knowledge of the language).

I think this work is perfect in its our way, highlighting the difficulties in narrative texts. This is a somewhat strucured way of dicussing the issue and calling attention to it. This is refreshing in a way- no statistical calculations, probabilities and significances; just real word experiences and comments of those involved. At the same time we cannot call it a 'scientific' paper, and the conclusion can be the same as the induction: hoarding is a complex, multifaced condition, hard to tackle and needing multidisciplinary approach.

Considering this point, the paper should not be published; on the other hand, it should be published because of the paucitiy of hoarding literature, because it calls attention to this difficult multiangle problem may be evoking further comments and suggestions including psychiatric treatment and authorities' careful intervention of management.

For the latter reason and hope, I suggest to publisg the paper.

Reviewer #2: The authors provide a sound argument. The study also includes a diverse group of professionals to appreciate the challenges they face from their perspective. Given that the participants were all recruited from the attendees of a regular forum to discuss hoarding, it is interesting to know that there were varied levels of understanding of hoarding and ways to manage the situation. If possible, the researchers can follow up with a debriefing or presentation at these forums to further emphasize the need for multi-disciplinary action and a consensus on the lengths and breaths of hoarding.

Reviewer #3: Thank you to the authors for the opportunity to review this paper. The need for this work is evident and provides greater awareness for understanding how we can support this under-represented population. The paper is overall well written and provides a compelling picture of the difficulties in working with hoarding populations. I believe it will be very useful in improving our care for hoarding within the community. There are some minor changes and recommendations made below. I hope they prove to be useful.

- In the introduction, the first sentence should be supported with a reference, perhaps Timpano et al. (2013)

Timpano, K. R., Broman-Fulks, J. J., Glaesmer, H., Exner, C., Rief, W., Olatunji, B. O., ... & Schmidt, N. B. (2013). A taxometric exploration of the latent structure of hoarding. Psychological assessment, 25(1), 194.

- Second sentence should have a comma after “when this becomes extreme”

- In the introduction, it states that early intervention is less likely, possibly related to reduced insight and the stigmatising nature of the condition. Beyond this, it could be mentioned that the criteria for hoarding (in particular the clutter criterion) usually requires decades of accumulation of items before it is met. As such, many individuals distressed by object attachment would not meet criterion until hoarding behaviours become extreme, necessitating multiagency support.

- There could be a greater rationale for the necessity of MDTs in working with hoarding populations. For example, different professionals are involved due to the common fire risks, hygiene concerns, risk of losing housing, psychological treatment, safety mitigation, etc.

- Typo on page 4: diverse backgroundS

- The results were very interesting in that many of the opinions of the professionals are well-aligned with emerging literature in hoarding populations. For example, professionals noted that hoarding groups had specific complexity including complex social needs, a history of trauma, and use possessions to replace social interaction. These are directly aligned with emerging social impairment models of hoarding (e.g., Chen et al., 2022), which could be noted within the discussion. This may suggest that professionals struggle with these issues because they are consistently observed social difficulties within this population, and therefore these results support the research within the hoarding space, which has suggested creating interventions that target these social difficulties too (e.g., David et al., 2022).

Chen, W., McDonald, S., Wearne, T., & Grisham, J. R. (2022). Interpersonal functioning in hoarding: An investigation of the link between hoarding symptoms and social support, social anhedonia, and social rewards. Journal of Affective Disorders Reports, 8, 100313.

David, J., Crone, C., & Norberg, M. M. (2022). A critical review of cognitive behavioural therapy for hoarding disorder: How can we improve outcomes?. Clinical psychology & psychotherapy, 29(2), 469-488.

- The discussion was interesting and synthesised the results well, including providing interesting recommendations for the field. Suggestions are as follows:

- On page 16, it could be useful to include references of papers outlining biopsychosocial understandings of hoarding on page 16, e.g., Mathes et al. (2022)

Mathes, B. M., Timpano, K. R., Raines, A. M., & Schmidt, N. B. (2020). Attachment theory and hoarding disorder: A review and theoretical integration. Behaviour research and therapy, 125, 103549.

- Regarding evidence-based interventions, Buried in Treasures is a standard intervention that could be recommended within the discussion. There is evidence to support it’s utility as a gold-standard CBT based protocol for hoarding, and has been found to be effective when conducted by mental health professionals, in support groups, or as a self-help program. Perhaps this could be mentioned in pages 16-17 when discussing how to involve and target psychological factors in hoarding.

- On page 17, the authors discuss the need to provide clear or dedicated care pathways to support individuals who hoard. As a broad stroke recommendation, findings (including from the literature on harm reduction approaches; Tompkins, 2011) may suggest that targeting safety and harm minimisation is usually a top priority, followed by prioritising psychological intervention. Notably, once bare minimum harm minimisation is met, harm reduction approaches take up to 2 years, suggesting that multiple concurrent intervention may be useful too, including further harm reduction with psychological intervention.

Tompkins, M. A. (2011). Working with families of people who hoard: a harm reduction approach. Journal of clinical psychology, 67(5), 497-506.

- Typo page 17: “dedicated care pathways TO support”

- This may not be a question that can be answered, but one that comes up in the context of this research, so I’m curious about the author’s thoughts. Regarding recommendations, it is useful to suggest care pathways that incorporate multi-agency collaboration. However, there are limitations to doing this, including agencies being under-resourced in being able to train professionals on hoarding specifically, and hoarding not being recognised as a top priority by government policies, and thus not attracting significant grant funding to improve research as well. Would the authors have additionally suggestions about how this could be targeted over time?

6. PLOS authors have the option to publish the peer review history of their article (what does this mean? ). If published, this will include your full peer review and any attached files.

**Do you want your identity to be public for this peer review?** For information about this choice, including consent withdrawal, please see our Privacy Policy .

Reviewer #1: **Yes: ** dr Anna Szucs

Reviewer #2: **Yes: ** Araba Aseye Ahiabu

Reviewer #3: **Yes: ** Wenting Chen

---

## [Author Response · Author response to Decision Letter 0]

13 Nov 2024

We would like to take this opportunity to thank the reviewers for their detailed and insightful comments. We have worked to address these individually and have provided responses to reviewer comments with details of each of the amendments made. Further, we provide responses to the additional journal requirements noted in your email dated 6th September 2024, including updated information pertaining to the Data Availability Statement (see “Response to Reviewers” document). We hope the amendments are sufficient and believe the manuscript has been significantly strengthened as a result, increasing its contribution to literature in this field.

---

## [Decision Letter · Decision Letter 1]

11 Mar 2025

PONE-D-23-41611R1“Falling between the cracks”: Investigating the competing challenges experienced by professionals working with people who hoard.PLOS ONE

Dear Dr. Kaminskiy,

Thank you for submitting your manuscript to PLOS ONE. After careful consideration, we feel that it has merit but does not fully meet PLOS ONE’s publication criteria as it currently stands. Therefore, we invite you to submit a revised version of the manuscript that addresses the points raised during the review process.

We look forward to receiving your revised manuscript.

Kind regards,

Rayees Farooq

Academic Editor

PLOS ONE

Journal Requirements:

Reviewers' comments:

Reviewer's Responses to Questions

**Comments to the Author**

1. If the authors have adequately addressed your comments raised in a previous round of review and you feel that this manuscript is now acceptable for publication, you may indicate that here to bypass the “Comments to the Author” section, enter your conflict of interest statement in the “Confidential to Editor” section, and submit your "Accept" recommendation.

Reviewer #1: All comments have been addressed

Reviewer #3: All comments have been addressed

2. Is the manuscript technically sound, and do the data support the conclusions?

Reviewer #1: Yes

Reviewer #3: Yes

3. Has the statistical analysis been performed appropriately and rigorously? 

Reviewer #1: N/A

Reviewer #3: Yes

4. Have the authors made all data underlying the findings in their manuscript fully available?

Reviewer #1: No

Reviewer #3: Yes

5. Is the manuscript presented in an intelligible fashion and written in standard English?

Reviewer #1: No

Reviewer #3: Yes

6. Review Comments to the Author

Reviewer #1: ’Falling between the cracks. Investigating the competing challenges experienced by professionals working with people who hoard.”

This remains a stopgap paper, dealing with the extremely severe and disregarded issue of hoarding disorder.

The authors review the difficulties of working with people who hoard and do not attend to the important question - what kind of disorder is it, and consequently, how could we treat it. At the same time, indeed, they stand to their word given in the title- dealing just with the challenges of managing the problem; however, it would be nice to know some more about the condition itself- maybe a too ambitious expectation at this point.

One have to esteem the authors’ efforts in addressing this burning but neglected field. They have followed reviewers’ suggestions, just a few typos remained.

For the latter reason I suggest ’minor revision’; but I propose to publish the paper in the hope of evoking further ones on the nature and treatment of hoarding.

Reviewer #3: Thank you to the authors for their thorough response to the comments. This paper will prove to be an exciting addition to the field.

7. PLOS authors have the option to publish the peer review history of their article (what does this mean? ). If published, this will include your full peer review and any attached files.

**Do you want your identity to be public for this peer review?** For information about this choice, including consent withdrawal, please see our Privacy Policy .

Reviewer #1: No

Reviewer #3: **Yes: ** Wenting Chen

---

## [Author Response · Author response to Decision Letter 1]

19 Mar 2025

Responses to reviewer:

We thank the reviewer for the time taken to re-read our paper and for the thoughtful comments. We wholeheartedly agree that hoarding disorder is a severe and neglected topic and thank the reviewer for the positive feedback. We have carefully checked the manuscript and corrected typos throughout. Thank you.

Responses to the editor:

We have double-checked the references and can confirm the reference list is complete and all sources have a corresponding citation in the text. We have checked and to the best of our knowledge, none of the references have subsequently been retracted.

---

## [Editor Report · Decision Letter 2]

8 Apr 2025

“Falling between the cracks”: Investigating the competing challenges experienced by professionals working with people who hoard.

PONE-D-23-41611R2

Dear Dr. Kaminskiy,

We’re pleased to inform you that your manuscript has been judged scientifically suitable for publication and will be formally accepted for publication once it meets all outstanding technical requirements.

Kind regards,

Rayees Farooq

Academic Editor

PLOS ONE
---

## [Editor Report · Acceptance letter]

PONE-D-23-41611R2

PLOS ONE

Dear Dr. Kaminskiy,

I'm pleased to inform you that your manuscript has been deemed suitable for publication in PLOS ONE. Congratulations! Your manuscript is now being handed over to our production team.

Kind regards,

on behalf of

Dr. Rayees Farooq

Academic Editor

PLOS ONE